# Design and synthesis of Nrf2-derived hydrocarbon stapled peptides for the disruption of protein-DNA-interactions

Bianca Wiedemann[1☯], Dominic Kamps[1,2☯], Laura Depta[1☯], Jörn Weisner[1], Jana Cvetreznik[1], Stefano Tomassi[3], Sascha Gentz[4], Jan-Erik Hoffmann[4], Matthias P. Müller[1], Oliver Koch[5], Leif Dehmelt[1,2], Daniel Rauh[1] *

1 Faculty of Chemistry and Chemical Biology, TU Dortmund University and Drug Discovery Hub Dortmund (DDHD), Zentrum für Integrierte Wirkstofforschung (ZIW), Dortmund, Germany, 2 Department of Systemic Cell Biology, Max Planck Institute of Molecular Physiology, Dortmund, Germany, 3 Department of Pharmacy, University of Naples "Federico II", Napoli, Italy, 4 Protein Chemistry Facility, Max Planck Institute of Molecular Physiology, Dortmund, Germany, 5 Institute of Pharmaceutical and Medicinal Chemistry and German Center of Infection Research, Münster, Germany

☯ These authors contributed equally to this work.
* daniel.rauh@tu-dortmund.de

**Data Availability Statement:** All relevant data are within the paper and its Supporting Information files.

**Funding:** This work was co funded by the German Federal ministry for Education and Research

## Abstract

Misregulation and mutations of the transcription factor Nrf2 are involved in the development of a variety of human diseases. In this study, we employed the technology of stapled peptides to address a protein-DNA-complex and designed a set of Nrf2-based derivatives. Varying the length and position of the hydrocarbon staple, we chose the best peptide for further evaluation in both fixed and living cells. Peptide **4** revealed significant enrichment within the nucleus compared to its linear counterpart **5**, indicating potent binding to DNA. Our studies suggest that these molecules offer an interesting strategy to target activated Nrf2 in cancer cells.

## Introduction

The transcription factor Nrf2 (nuclear factor erythroid 2-related factor 2) belongs to the cap'n'-collar (CNC) basic-region leucine zipper (bZIP) transcription factor family and consists of a total of seven domains [1, 2]. Nrf2 is a central component of the cellular detoxification machinery and mediates the transcription of approximately 250 genes. Those genes encode a variety of enzymes involved in the metabolism of protein degradation and the regulation of inflammatory responses such as Peroxiredoxin-1 (PRDX1), glutathione S-transferase (GST) and multidrug resistance-associated protein 1 (MRP). This allows for a multifaceted cellular response to endogenous and exogenous oxidative stress that ensures maintenance of homeostasis [3].

The versatile cytoprotective properties of the Keap1/Nrf2 pathway serve as a protective function against diseases underlying oxidative stress or inflammatory mechanisms. Overactivation of Nrf2 has also been observed in tumor cells [4]. In this case, the proteasomal degradation of Nrf2 is suppressed by genetic alterations in Nrf2 and Keap1, but also by oncogenes

(NGFNPlus and e:Med) (Grant No. BMBF 01GS08104, 01ZX1303C), the Deutsche Forschungsgemeinschaft (DFG), the German federal state North Rhine Westphalia (NRW) and the European Union (European Regional Development Fund: Invest In Your Future) (EFRE-800400), NEGECA (PerMed NRW) and EMODI. The funders had no role in study design, data collection and analysis, decision to publish, or preparation of the manuscript.

**Competing interests:** The authors have declared that no competing interests exist.

such as KRASG12D or BRAFV619E, and the protective functions of the resulting gene products are exploited for the metabolization of chemotherapeutic agents or for the defense against radicals produced as a result of radiation therapy. This leads to increased tumor progression, metastasis and therapy resistance in cancer, making Nrf2 an attractive therapeutic target [2, 5–9]. Unfortunately, proteins like Nrf2 are intrinsically difficult to target due to their disordered structure in solution and their lack of known binding pockets responsible for their activity [10]. For several transcription factors (Tfs), addressing protein-protein interaction interfaces has proven to be a successful strategy [11]. However, none of these approaches have succeeded in downregulation of Nrf2 to target its detrimental role in cancer development [12, 13]. A promising alternative strategy is to target the protein-DNA-interactions (PDI) [14]. Multiple binders of DNA have been known for several years, but a lack of specificity prevented their use for therapeutic approaches [15]. Thus, the development of molecules that are able to bind highly specifically to certain DNA sequences while retaining cellular permeability is a novel, challenging, and promising approach to overcome the limitations outlined above [14]. Within the last decade, stabilized helical peptides have been successfully employed to target so called "undruggable" proteins and especially protein-protein-interactions [16–28]. Here, we report the synthesis of α-helical hydrocarbon stapled peptides derived from Nrf2 for the inhibition of its interaction with DNA.

## Materials and methods

Unless otherwise noted, all reagents and solvents were purchased from Acros, Fluka, Sigma-Aldrich, Merck or Okeanos Technology Co., Ltd. and used without further purification. Dry solvents were purchased as anhydrous reagents from commercial suppliers. Compounds were purified by a preparative Agilent HPLC system (1200 series) with a VP 125/21 Nucleodur C18 column from Macherey-Nagel and monitored by UV at $\lambda$ = 210 nm and 280 nm. All final compounds were purified to >95% purity as determined by high-performance liquid chromatography (HPLC).

### Homology modeling

The Uniprot protein sequence Q16236 (NF2L2_HUMAN) was used as starting point. A blast search (https://blast.ncbi.nlm.nih.gov/Blast.cgi) using Q16236 revealed the complex structure pdb 1skn as suitable template structure. Homology modeling was performed using the "homology modeling" tool implemented in MOE 2013. The option "include selected atoms as environment for induced fit option" was enabled to transfer the DNA to the homology model. A comparison to a solution nmr structure of NRF2 (pdb 2lz1) showing the lower helix bundle reveals a perfect structural overlap of corresponding residues indicating a high quality model [29].

### Peptide synthesis

The peptides were prepared on solid support and via Fmoc-based synthetic strategy incorporating non-natural amino acids ((S)-N-Fmoc-2-(4′-pentenyl)alanine (S5), (R)-N-Fmoc-2-(4′-pentenyl)alanine (R5), (R)-N-Fmoc-2-(7′octenyl)alanine) (R8) at the respective stapling positions. All peptides were prepared bearing a C-terminal amide on Rink amide or MBHA Rink amide resin. PyCloK in DMF was used as a coupling reagent solution. Amino acid building blocks were coupled in four-fold excess respectively, while non-natural aa were used in 3-fold excess. For the formation of the macrocycle via ring closing metathesis, 1st gen. Grubbs catalyst in dry DCE was used while $N_2$ was guided through the solution. Piperidine (25%) in DMF was used as the deprotection solution. Peptides were either modified with a

fluorescent label or capped at the N-terminus. Implementation of FITC as label needed an additional linker between the N-terminal amino acid and the label to prevent undesired side reactions. All peptides were cleaved using a cocktail of 95% trifluoroacetic acid (TFA) and 5% triisopropylsilane (TIS), subsequently precipitated with diethylether, and purified via HPLC.

## CD measurements

CD measurements were carried out on a JASCO J-715 spectrometer using water as the solvent for all measurements. The peptides were solved in water to a concentration of 20 μM. The linear peptide **5** was used as a control compound. The helical fraction for peptides **1**–**5** was calculated from the observed ellipticity $[\theta]_{obs}$ at 222 nm using established methods [30].

## Fluorescence polarization assay

For DNA hybridization, equimolar amounts of fluorescein- or 6FAM-labeled sense and unlabeled antisense oligonucleotides were mixed in 1X annealing buffer (10 mM Tris pH 7.5, 100 mM NaCl, 1 mM EDTA) to yield a final concentration of 20 μM of dsDNA. Subsequently, mixtures were heated to 95˚C for 10 min followed by cooling to 20˚C at a rate of 1˚C per minute. Afterwards, dsDNA probes were diluted to 10 nm in FP assay buffer (50 mM Tris/HCl pH 7.5, 100 mM NaCl, 2 mM MgCl2, 0.5 mM EDTA, 10 μg/mL BSA, 0.1 mM DTT, 0.01% Tween-20) and mixed with serial dilutions of peptides **1**–**6** (25 μM—6.1 nm), that were generated with the Echo 520 liquid handler (Labcyte Inc.), in a total volume of 10.2 μL per well in black 384-well small volume microplates (Greiner Bio-One). The microplates were shaken at 1500 rpm for 30 s, centrifuged at 200 x g for 60 s and subsequently incubated for 60 min in a humidified, dark chamber at room temperature. Finally, the fluorescence polarization was analyzed on an Infinite® M1000 microplate reader (Tecan) using the appropriate settings for fluorescein/6FAM, i.e. an excitation wavelength of 470 ± 2.5 nm and an emission wavelength of 520 ± 7.5 nm.

Each concentration was analyzed in quadruplicates and resulting mean fluorescence polarization and standard deviation was fit to a four parameter logistic model with Origin (OriginLab, Northampton, MA) using the following equation:

$$y = \left( \frac{A_1 - A_2}{1 + \left(\frac{x}{x_0}\right)^p} \right) + A_2$$

Each experiment was performed in three independent replicates.

## Electrophoretic mobility shift assay

DNA hybridization was performed as described above. Afterwards, 1 pmol of the dsDNA probe was incubated with the respective excess of peptide in 1X assay buffer (10 mM Tris pH 7.5, 100 mM KCl, 2 mM MgCl2, 0.5 mM EDTA, 10 μg/mL BSA, 0.1 mM DTT, 2.5% glycerol) in a final volume of 10 μL and incubated for 30 min at room temperature. Prior to gel electrophoresis, samples were mixed with 2 μL of 6X loading buffer (60 mM Tris pH 7.5, 15% Ficoll 400) and the reaction mixtures were then loaded onto hand-cast, native 15% TBE polyacrylamide gels. Separation was performed for 45 min at a current of 20 mA per gel in a cold room at 4˚C and the gels were analyzed on a Typhoon FLA 9500 laser scanner (GE Healthcare) using the 473 nm laser for excitation and a 510 nm long-pass filter (LPB (510LP)) for emission of fluorescein and FAM, respectively.

## Cell culture and pharmacological treatments

HeLa cells (ATCC CCL-2) were cultured in Dulbecco's Modified Eagle Medium (DMEM) supplemented with 10% FBS (Pan Biotech), 2 mM L-glutamine, 100 U/ml penicillin and 100 μg/ml streptomycin at 37°C and 5% $CO_2$.

For fixing and permeabilization, HeLa cells were incubated with pre-warmed 4% formaldehyde for 20 min at 37°C, followed by permeabilization with 0.25% Triton-X 100 for 20 min at room temperature. For the evaluation of the effect of proteases, cOmplete protease inhibitor mix (1X) 500 μl was added to the fixed cells for 30 min. Peptides were incubated at 1 μM for 3 h in the presence of the protease inhibitor mix, followed by 3 min incubation with 1 μM of the nucleic acid marker DAPI, then extensively washed with DPBS and subjected to wide-field imaging.

To investigate the cell permeability of peptide **4**, living HeLa cells were washed twice and incubated for 5 min with DPBS + Mg/Ca, (100 mg/ml $CaCl_2$, 133 mg/ml $MgCl_2$) in a $CO_2$ incubator. Cells were subsequently incubated in the presence of 10 μM of peptide **4** in DPBS + Mg/Cl for 5 min, followed by two washes with DPBS + Mg/Cl and final addition of HEPES-buffered imaging medium containing 0.5 μM propidium iodide.

Imaging was performed using an Olympus IX81 microscope with 20x and 60x UPLSAPO objectives and standard fluorescence filter sets.

## Results and discussion

Using the crystal structure of the Tf Skn1 (PDB 1skn), [31] we created a homology model of the DNA binding domain of Nrf2 for the design of our peptides (S1 Fig in S1 File). Critical evaluation of this model guided the classification of the Nrf2 amino acids (aa) into four groups (Fig 1): a) aa from the a-helical segment docking in the DNA major groove and forming hydrogen bonds with either the DNA bases or the phosphate backbone; b) aa within the sequence of a) but facing away from the DNA; c) aa within reach of the DNA that do not

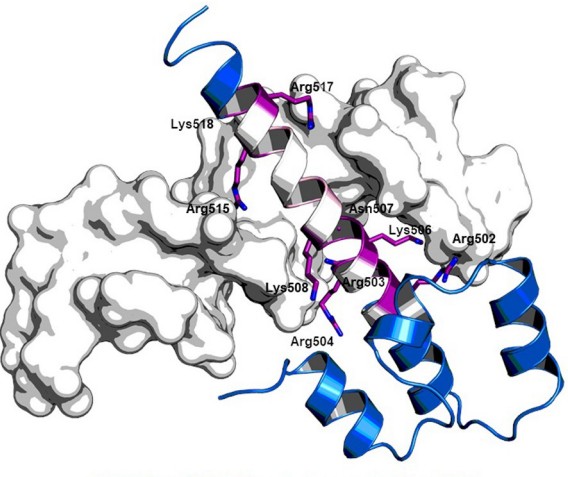

**Fig 1. Homology model of the helix-loop-helix domain of Nrf2 based on Skn1 (PDB 1skn) bound to DNA.** Amino acids (aa) of group a) which interact with the DNA are colored in purple; aa from group b) which are suitable to introduce a staple are colored in white; aa from group c) which are suitable for modification are colored in rose; aa from group d) which have no interaction with the DNA are colored in blue. Bottom: aa sequence used for the design of stapled peptides. The aa are divided into groups a)-d) according to color code.

interact with it; d) aa 452–500 and 519–525 that are too remote from the DNA to allow interactions (these aa were excluded from further consideration for the design of the stapled peptides).

For our first design, we left aa from groups a) and c) unchanged and engaged those from group b) for potential all-hydrocarbon stapling. We further varied the position and length of the staples to elucidate the best position within the sequence for induction of an α-helical character (the general synthesis scheme is depicted in Fig 2A). We chose positions 505/509 (1) and 501/505 (2) for an i,i+4 stapling system, 509/512 (3) for an i,i+3 stapling system and 509/516 (4) for an i,i+7 stapling system (Fig 2B).

Subsequent CD measurements were used to determine the α-helical content of the respective peptides (S2 Fig and S3 Table in S1 File). While the non-stapled linear peptide 5 had an α-helical content of 6%, this value was significantly increased upon introduction of all-hydrocarbon staples. Indeed, peptide 3 exhibited the highest α-helical content with 36%, followed by peptide 1 (20%) and peptide 4 (18%). Conversely, peptide 2 revealed an α-helicity of 9% which is in the same range of the linear peptide 5 (α-helicity = 6%) thus showing that placing an all-hydrocarbon staple between 501/505 positions is detrimental for the α-helical induction.

To evaluate the binding affinities of these peptides to the Nrf2 binding site, the double stranded DNA sequence responsive to Nrf2, we performed fluorescence polarization

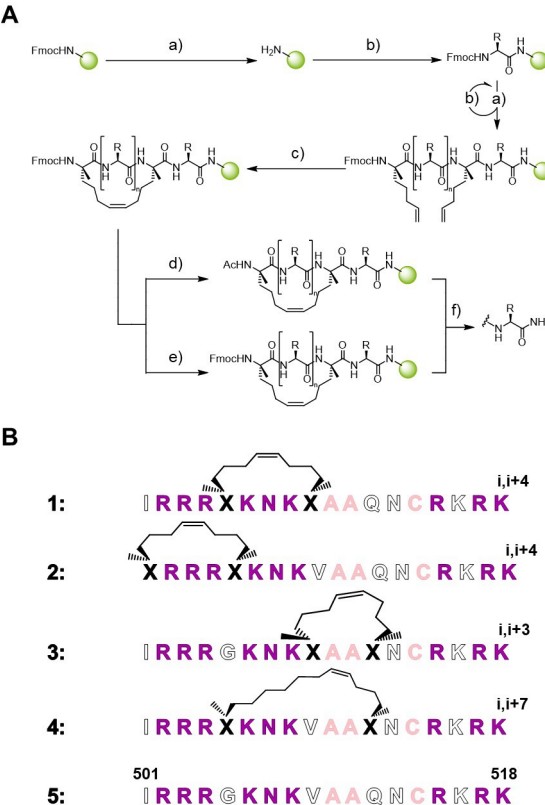

**Fig 2.** A: Synthesis scheme for stapled peptides. a) 25% piperidine in DMF, 2 x 15 min, rt; b) 4 eq. PyClocK, 4 eq Fmoc-NH-aa-COOH, 4–8 eq DIPEA in DMF, 2 h, rt; for non-natural aa: 3–4 eq. PyClocK, 3 eq. Fmoc-NH-aa-COOH, 4–6 eq. DIPEA in DMF, 2 h, rt; c) 10 mol% 1st gen. Grubbs catalyst in DCE, 3 x 1 h, rt; d) 4 eq. Ac2O, 8 eq. DIPEA, DMF, 45 min, rt; e) 5 eq. FITC, 10 eq. DIPEA, DMF, 24 h, rt; f) excess TFA with 5% TIS, 2 h, rt. B: Sequence of stapled peptides 1–5 derived from Nrf2. Derivative 1 (i,i+4 stapling); derivative 2 (i,i+4 stapling); derivative 3 (i,i+3 stapling); derivative 4 (i,i+7 stapling). Color coding according to Fig 1.

experiments (Fig 3 and S5 Fig in S1 File) and electrophoretic mobility shift assays (S6-S8 Figs in S1 File).

We used three FAM-labelled double stranded DNA sequences in these experiments: MARE23 as a known Nrf2:sMaf binding sequence, [32] a scrambled version of this sequence (MARE23_scr) and an additional randomized DNA sequence as controls.

The FP measurements show a higher binding affinity for the stapled peptides **1** ($K_D$ = 1.6 ± 0.04 µM), **2** ($K_D$ = 3.9 ± 0.2 µM), **3** ($K_D$ = 1.5 ± 0.2 µM) and **4** ($K_D$ = 1.2 ± 0.3 µM) to MARE23 compared to the linear peptide **5** ($K_D$ = 7.9 ± 0.1 µM) and the scrambled peptide **6** ($K_D$ = 8.1 ± 0.1 µM). (Fig 3) Additionally, the stapled peptide **4** is able to induce a mobility shift of DNA at a lower concentration than the unstapled peptide **5** (S6-S8 Figs in S1 File). However, the FP measurements and electrophoretic mobility shift studies indicated that the stapled peptide also binds to the randomized DNA sequences used as negative controls thus indicating the need for further optimization of the peptides to obtain higher sequence selectivity.

Modification of peptide **4** and the non-stapled peptide **5** at the N-terminus with a fluorescent label rendered the peptides detectable via fluorescence microscopy. Since direct modification with FITC would cause the terminal aa to undergo Edman degradation, a suitable linker was incorporated. We chose γ-amino-butyric-acid (GABA) for this purpose, thereby providing a suitable spacing function between the fluorophore and the studied peptide sequence.

To analyze their potential interaction with cellular DNA, we incubated peptides FITC-**4** and FITC-**5** with fixed, permeabilized HeLa cells. As shown in Fig 4, peptide FITC-**4** was selectively enriched in the DNA-rich nucleus. In contrast, only small amounts of linear peptide FITC-**5** were detectable in cells and no enrichment in the nucleus could be observed for peptide FITC-**5**, suggesting that nuclear enrichment of the peptides is dependent on stapling and the resulting higher affinity towards DNA.

These data corroborate the results of the fluorescence polarization assay, indicating a higher binding affinity of the stapled peptide **4** compared to the linear peptide **5**. Similar results were obtained conducting the same cellular microscopy experiments but in the presence of protease inhibitors (S9 Fig in S1 File). This shows that the differences in cell permeability and subcellular localization observed for peptides FITC-**4** and FITC-**5** were most probably not caused by selective degradation of the unstapled peptide by residual protease activity (S9 Fig in S1 File). In addition to binding nuclei in fixed and permeabilized cells, the FITC labeled peptide FITC-

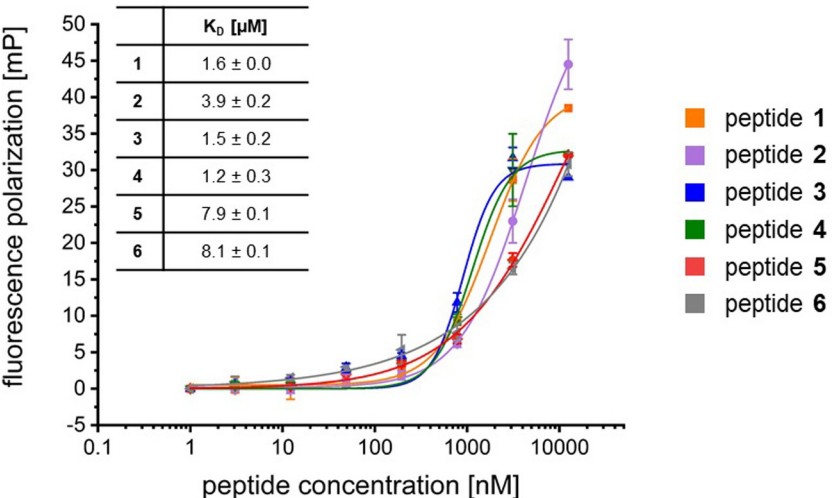

**Fig 3. Fluorescence polarization assay of peptides 1–6 with 6FAM-labeled MARE23 sequence.**

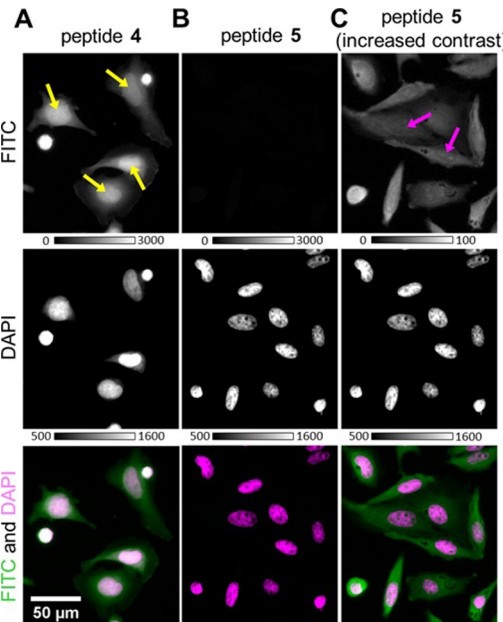

**Fig 4. Enrichment of peptide 4 in DNA-rich nuclei.** A-C: Fixed and permeabilized HeLa cells were incubated with peptides 4 and 5 and imaged using wide-field fluorescence microscopy. Representative images of FITC-labeled peptides (top panels), DAPI (middle panels), and the combined images (bottom panels) are shown. A, B: Images showing peptides 4 and 5, respectively, with identical exposure times and image scaling. C: Images of peptide 5 with 30-fold increased image contrast to visualize weak fluorescence of FITC signals.

**4** also showed high membrane permeability in living cells (Fig 5). After incubation in the presence of 10 µM of peptide FITC-**4** and subsequent washing, cytosolic and nuclear localization of the peptide was observed, with increased signal in nuclear substructures that resemble nucleoli (Fig 5B).

Addition of the membrane impermeable dye propidium iodide confirmed that uptake of peptide FITC-4 was not simply due to defects in the plasma membrane and that the majority of peptide-containing cells were alive. Based on these findings, we will focus on another round of improved stapled peptides guided by our presented design and results in the future, maintaining the stapling system featured by peptide 4 (i,i+7) and focusing on the modification of aa of the aforementioned group c) (see above).

## Conclusion

In summary, we successfully designed and synthesized a library of Nrf2 derived stapled peptides to address nucleic acids in cells. For one of these peptides, we successfully demonstrate nucleic acid binding and efficient uptake in living cells. The stapled peptides were designed to bind the major groove of a specific DNA sequence known as the antioxidant response element. In cells, compared to its linear counterpart, the stapled derivative exhibited elevated binding, in particular to nuclear areas. Indicated by the electrophoretic mobility shift experiments and the fluorescence polarization assay, the stapled peptides **1**–**4** show an improved binding affinity for DNA than unstapled peptide **5** and the scrambled peptide **6**, but still require optimization in terms of the selectivity towards different DNA sequences. Recently, Simov *et al.* reported a series of peptides that demonstrate high affinity and selective binding to the Antioxidant Response Element (ARE) DNA and thereby displace NRF2 from its promoter, indicating the general applicability of this approach [33]. Further studies for the optimization of sequence

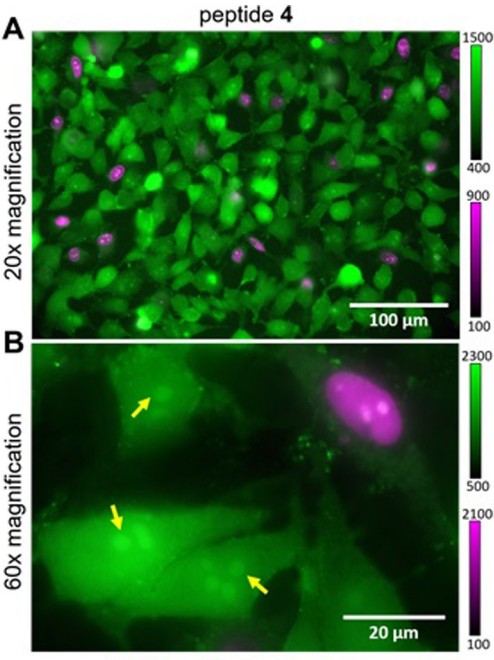

**Fig 5. Cell permeability of peptide 4 in living cells.** A: Representative combined image of HeLa cells treated with FITC labeled peptide 4 under serum- free conditions. Propidium iodide was added to stain a small subpopulation of necrotic and late apoptotic cells, which are permeable due to membrane defects. B: Localization of peptide 4 in the cytosol and enrichment in nuclear substructures (yellow arrows).

specificity of our peptide library are underway. For this purpose, replacement of critical amino acids with, e.g., unnatural mimetics could render the peptides more potent.

The approach described above will, however, not only be applicable to DNA but also represents a promising tool to address RNA. Folding of RNAs into secondary structures like loops, pseudo-knots, and quadruplexes are frequently found in RNA and can be associated with a gain of function for these molecules [34–36]. Tight hydrogen bonds within these folded nucleic acids render therapeutic means like RNAi futile [34, 36]. Small molecules are also prone to fail in such cases due to the large shallow surface areas that these RNAs offer for binding. Thus, employing stapled peptides to address these nucleic acid structures might yield a better understanding of their function to ultimately facilitate the treatment of associated diseases. However, due to the overall negative charge of nucleic acids, great care must be taken to ensure that nonspecific electrostatic interactions do not interfere with the desired specific binding, requiring careful design and evaluation of the stapled peptide library.

## Supporting information

**S1 File.**
(DOCX)

**S1 Raw images.**
(PDF)

## Acknowledgments

We are grateful to Tom Grossmann and Adrian Glas for helpful discussions.

## Author Contributions

**Conceptualization:** Daniel Rauh.

**Data curation:** Bianca Wiedemann.

**Formal analysis:** Bianca Wiedemann, Dominic Kamps, Laura Depta, Jörn Weisner, Jana Cvetreznik, Stefano Tomassi, Sascha Gentz, Jan-Erik Hoffmann, Matthias P. Müller, Oliver Koch.

**Funding acquisition:** Daniel Rauh.

**Investigation:** Bianca Wiedemann, Dominic Kamps, Laura Depta, Jörn Weisner, Stefano Tomassi, Leif Dehmelt.

**Project administration:** Daniel Rauh.

**Resources:** Leif Dehmelt, Daniel Rauh.

**Validation:** Bianca Wiedemann, Dominic Kamps, Laura Depta.

**Visualization:** Bianca Wiedemann, Dominic Kamps, Laura Depta.

**Writing – original draft:** Bianca Wiedemann.

**Writing – review & editing:** Dominic Kamps, Laura Depta, Jörn Weisner, Matthias P. Müller, Leif Dehmelt, Daniel Rauh.

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
