## [Decision Letter · Decision Letter 0]

2 Apr 2022

PONE-D-22-01699Design and Synthesis of Nrf2-Derived Hydrocarbon Stapled Peptides for the Disruption of Protein-DNA-InteractionsPLOS ONE

Dear Dr. Rauh,

Thank you for submitting your manuscript to PLOS ONE. After careful consideration also based on my own reading, we feel that it has scientific merit but does not fully meet PLOS ONE’s publication criteria as it currently stands. Therefore, we invite you to submit a revised version of the manuscript that addresses the points raised during the review process. Editor's suggestions1) Could the authors give more details about the homology modeling  procedure in the main text (now is reported in the SI)? It is a critical part of the work as it has inspired the design and synthesis of the peptide library.2) The authors  commented on the flexibility of their approach which could lead to agents able to interfere with RNA activity. I agree with their vision, but I suggest them to also mention the potential selectivity issues emerging from the direct targeting of nucleic acids with peptides.  Please submit your revised manuscript by May 17 2022 11:59PM. If you will need more time than this to complete your revisions, please reply to this message or contact the journal office at plosone@plos.org. Please include the following items when submitting your revised manuscript:A rebuttal letter that responds to each point raised by the academic editor and reviewer(s). You should upload this letter as a separate file labeled 'Response to Reviewers'.A marked-up copy of your manuscript that highlights changes made to the original version. You should upload this as a separate file labeled 'Revised Manuscript with Track Changes'.An unmarked version of your revised paper without tracked changes. You should upload this as a separate file labeled 'Manuscript'.

We look forward to receiving your revised manuscript.

Kind regards,

Alessio Lodola, PhD

Academic Editor

PLOS ONE

Journal Requirements:

“We are grateful to Tom Grossmann and Adrian Glas for helpful discussions. This work was co funded by the German Federal ministry for Education and Research (NGFNPlus and e:Med) (Grant No. BMBF 01GS08104, 01ZX1303C), the Deutsche Forschungsgemeinschaft (DFG), the German federal state North Rhine Westphalia (NRW) and the European Union (European Regional Development Fund: Invest In Your Future) (EFRE-800400), NEGECA (PerMed NRW) and EMODI.”

 “This work was co-funded by the German Federal ministry for Education and Research (NGFNPlus and e:Med) (Grant No. BMBF 01GS08104, 01ZX1303C), the Deutsche Forschungsgemeinschaft (DFG), the German federal state North Rhine Westphalia (NRW) and the European Union (European Regional Development Fund: Invest In Your Future) (EFRE-800400), NEGECA (PerMed NRW) and EMODI.

Reviewers' comments:

Reviewer's Responses to Questions

**Comments to the Author**

1. Is the manuscript technically sound, and do the data support the conclusions?

Reviewer #1: Yes

2. Has the statistical analysis been performed appropriately and rigorously? 

Reviewer #1: N/A

3. Have the authors made all data underlying the findings in their manuscript fully available?

Reviewer #1: Yes

4. Is the manuscript presented in an intelligible fashion and written in standard English?

Reviewer #1: Yes

5. Review Comments to the Author

Reviewer #1: The authors describe the design, synthesis and testing of stapled peptides meant to target a specific DNA sequence. The authors show convincingly that the peptides bind to DNA, albeit they are not specific to the targeted sequence but bind to other DNA sequences equally well.

The experiments are generally described and carried out well, thus the results are well worse publishing in PLOS one. (Note: I am not an expert in peptide synthesis, therefore I can’t comment on this part of the work and my comments relate only to the non-synthesis parts of the manuscript.)

However, the following minor comments should be addressed before publication:

1) The authors treat cells with a linear peptide and a stapled peptide. For the latter one, enrichment in the nucleus was observed and therefore the authors concluded that the enrichment is due to stapling. However, both peptides were used at the same concentration, even so the binding affinity for the stapled peptide for DNA is about 6-fold higher than for the linear one. The authors should therefore discuss if the observed enrichment in the nucleus could be rather driven by affinity than the stapling itself.

2) The result and discussion section ends rather sudden. In the last sentence, it says that the authors decided to embark in a second round of optimization. However, the optimization is not described in the paper. Is there something lacking in the manuscript or is the second round of optimization a future project?

3) Page 7, line 148, “DNA hybridization was performed as described previously”. Is there are reference lacking or do the authors rather mean “above” instead of “previously”?

4) Page 10, line 216, this should read “… is able to induce a mobility shift”

5) Next line: “… at a lower concentration in respect to the unstapled peptide 5 ..”. This is unclear. Should this read “… at a lower concentration than the unstapled peptide 5 …”?

6) Figure S1: Which colour hast the model and which colour the NMR structure?

6. PLOS authors have the option to publish the peer review history of their article (what does this mean?). If published, this will include your full peer review and any attached files.

Reviewer #1: **Yes: **Ruth Brenk

---

## [Author Response · Author response to Decision Letter 0]

11 Apr 2022

Response to Decision Letter: Wiedemann et al. 

[PONE-D-22-01699] 

Editor’s suggestions

1. Could the authors give more details about the homology modeling procedure in the main text (now is reported in the SI)? It is a critical part of the work as it has inspired the design and synthesis of the peptide library.

We agree that the homology model is an essential part of our presented work. We now include the homology modeling procedure in the main text (line 103-111). 

2. The authors commented on the flexibility of their approach which could lead to agents able to interfere with RNA activity. I agree with their vision, but I suggest them to also mention the potential selectivity issues emerging from the direct targeting of nucleic acids with peptides. 

We thank the editor for this comment. We now address the selectivity issues in the conclusion. 

The text reads:” However, due to the overall negative charge of nucleic acids, great care must be taken to ensure that nonspecific electrostatic interactions do not interfere with the desired specific binding, requiring careful design and evaluation of the stapled peptide library.“

Reviewer #1: 

The authors describe the design, synthesis and testing of stapled peptides meant to target a specific DNA sequence. The authors show convincingly that the peptides bind to DNA, albeit they are not specific to the targeted sequence but bind to other DNA sequences equally well. The experiments are generally described and carried out well, thus the results are well worth publishing in PLOS one.

We are very grateful and thank the reviewer for the positive assessment of our manuscript.

1. The authors treat cells with a linear peptide and a stapled peptide. For the latter one, enrichment in the nucleus was observed and therefore the authors concluded that the enrichment is due to stapling. However, both peptides were used at the same concentration, even so the binding affinity for the stapled peptide for DNA is about 6-fold higher than for the linear one. The authors should therefore discuss if the observed enrichment in the nucleus could be rather driven by affinity than the stapling itself.

We thank the reviewer for the comment and agree that the wording needs to be more precise to clarify this. What we meant to say is that stapling increases the affinity of the peptide and thereby leads to stronger binding and enrichment in the nucleus. 

The manuscript now reads:” In contrast, only small amounts of linear peptide FITC-5 were detectable in cells and no enrichment in the nucleus could be observed for peptide FITC-5, suggesting that nuclear enrichment of the peptides is dependent on stapling and the resulting higher affinity towards DNA.”

2. The result and discussion section ends rather sudden. In the last sentence, it says that the authors decided to embark in a second round of optimization. However, the optimization is not described in the paper. Is there something lacking in the manuscript or is the second round of optimization a future project?

This point is well taken. We will focus on another round of improved stapled peptides guided by our presented design and results in the future. 

To clarify this, we rephrased the sentence:” Based on these findings, we will focus on another round of improved stapled peptides guided by our presented design and results in the future, maintaining the stapling system featured by peptide 4 (i,i+7) and focusing on the modification of aa of the aforementioned group c) (see above).

3. Page 7, line 148, “DNA hybridization was performed as described previously”. Is there are reference lacking or do the authors rather mean “above” instead of “previously”?

We corrected for this. The text now reads:” DNA hybridization was performed as described above.”

4. Page 10, line 216, this should read “… is able to induce a mobility shift”

We corrected for this. 

5. Next line: “… at a lower concentration in respect to the unstapled peptide 5 ..”. This is unclear. Should this read “… at a lower concentration than the unstapled peptide 5 …”?

We corrected for this. The sentence now reads: “Additionally, the stapled peptide 4 is able to induce a mobility shift of DNA at a lower concentration than the unstapled peptide 5 (Fig. S6 8).”

6. Figure S1: Which colour has the model and which colour the NMR structure?

We added the colours to make this more clear.

---

## [Editor Report · Decision Letter 1]

13 Apr 2022

Design and Synthesis of Nrf2-Derived Hydrocarbon Stapled Peptides for the Disruption of Protein-DNA-Interactions

PONE-D-22-01699R1

Dear Dr. Rauh,

We’re pleased to inform you that your manuscript has been judged scientifically suitable for publication and will be formally accepted for publication once it meets all outstanding technical requirements.

Kind regards,

Alessio Lodola, PhD

Academic Editor

PLOS ONE
---

## [Editor Report · Acceptance letter]

4 May 2022

PONE-D-22-01699R1 

Design and Synthesis of Nrf2-Derived Hydrocarbon Stapled Peptides for the Disruption of Protein-DNA-Interactions 

Dear Dr. Rauh:

I'm pleased to inform you that your manuscript has been deemed suitable for publication in PLOS ONE. Congratulations! Your manuscript is now with our production department. 

Kind regards, 

on behalf of

Dr Alessio Lodola 

Academic Editor

PLOS ONE